# Dysregulation of the Amniotic PPARγ Pathway by Phthalates: Modulation of the Anti-Inflammatory Activity of PPARγ in Human Fetal Membranes

**DOI:** 10.3390/life12040544

**Published:** 2022-04-06

**Authors:** Audrey Antoine, Coraline De Sousa Do Outeiro, Coline Charnay, Corinne Belville, Fanny Henrioux, Denis Gallot, Loïc Blanchon, Régine Minet-Quinard, Vincent Sapin

**Affiliations:** 1iGReD, Team “Translational Approach to Epithelial Injury and Repair”, UMR6293 CNRS-U1103 INSERM, Université Clermont Auvergne, F-63000 Clermont-Ferrand, France; audrey.antoine@uca.fr (A.A.); coraline.de.sousa.do.outeiro@umontreal.ca (C.D.S.D.O.); coline.charnay@uca.fr (C.C.); corinne.belville@uca.fr (C.B.); fanny.henrioux@uca.fr (F.H.); dgallot@chu-clermontferrand.fr (D.G.); loic.blanchon@uca.fr (L.B.); regine.minet-quinard@uca.fr (R.M.-Q.); 2CHU Clermont-Ferrand, Obstetrics and Gynecology Department, F-63000 Clermont-Ferrand, France; 3CHU Clermont-Ferrand, Biochemistry and Molecular Genetics Department, F-63000 Clermont-Ferrand, France

**Keywords:** fetal membranes, PPARγ, phthalates, inflammation

## Abstract

Phthalates are reprotoxic pollutants that are omnipresent in the environment. Detectable in amniotic fluid, these compounds (with the most concentrated being mono-2-ethylhexyl phthalate (MEHP)) are in direct contact with fetal membranes (FMs). They can lead to the premature rupture of FMs by deregulating cellular and molecular pathways, such as, for example, the nuclear transcription factor peroxysome proliferator-activated receptor gamma (PPARγ) pathway. The objective was to study the impact of MEHP on the PPARγ pathway in FMs using amnion and choriodecidua across the three trimesters of pregnancy and the amniotic epithelial AV3 cell model by analyzing (i) PPARγ expression (mRNA and proteins) using RT-qPCR and Western blot assays; (ii) cytotoxicity and cell viability following MEHP treatment by lactate dehydrogenase (LDH) measurement and using Cell-counting Kit 8; and (iii) modulation by MEHP of PPARγ transcriptional activity (using a reporter gene assay) and PPARγ anti-inflammatory properties (by measuring IL6 and IL8 levels). PPARγ is expressed in the human amnion and choriodecidua during the three trimesters of pregnancy and in amniotic cells. In the AV3 cell line, MEHP is not cytotoxic and does not reduce cell viability, but it reduces PPARγ activity, here induced by a classical agonist without influencing its expression. MEHP also reduces PPARγ’s anti-inflammatory properties. In conclusion, PPARγ signaling is dysregulated by MEHP; this paves the way for future explorations to highlight the hypothesis of phthalates as an amniotic PPARγ disruptor that can explain the premature rupture of FMs.

## 1. Introduction

Preterm birth is the leading cause of neonatal mortality worldwide and can lead to several adverse maternal and newborn health effects. The preterm, prelabor rupture of membrane (pPROM) leads to 30% to 40% of preterm births worldwide and is defined as a rupture of the water pocket—called the fetal membrane (FM)—prior to 37 weeks of pregnancy [1]. FMs comprise two layers—the amnion and chorion—that are connected by a collagen-rich extracellular matrix, thus establishing a real interface between the mother and her future newborn [2]. FM rupture is a consequence of biochemical and histological changes during pregnancy [3,4]. These modifications mainly contribute to the formation of a weak zone called the zone of altered morphology (ZAM) at the cervix (site of rupture), which is characterized by a significant reduction in FM thickness [2]. This weakening is induced by the combination of exacerbated and programmed molecular and cellular events such as oxidative stress, apoptosis, senescence, extracellular matrix degradation, and, especially, sterile inflammation [5]. The level of proinflammatory proteins and cytokines such as IL-1β, IL-6, and TNFα increases in the amniotic fluid near term and is exacerbated in the case of pPROM [6,7], underlying that this phenomenon is important in the FMs’ physiological and pathological weakening. 

The peroxisome proliferator-activated receptor (PPAR) is a member of the nuclear receptor superfamily that binds with the peroxisome proliferator response element (PPRE) to regulate the transcription of the target genes involved in lipid metabolism, embryonic development, and inflammation, which is one of the key phenomena for FM rupture [8,9]. There are three isoforms for PPAR receptors—PPARα, β/δ, and γ—and each has its cell and tissue expression patterns and specificity of molecular and cellular action. Lappas et al. demonstrated that PPARγ can reduce the production of proinflammatory cytokines (IL-6, IL-8, and TNFα) induced by lipopolysaccharide in FMs and the placenta when the receptor is activated by its ligand: thiazolidinedione [10]. Due to its properties, this receptor may be an essential factor in the regulation of inflammation in FMs during pregnancy and can be involved in the dysregulation of this phenomenon near term. 

The esters of phthalic acid, also known as phthalates, are synthetic chemical compounds that are widely used as plasticizers in numerous products, including polyvinyl chloride (materials, food packaging, cosmetics, and medical devices) [11]. Among them, di-2-ethylhexyl phthalate (DEHP) is the most present in the environment because of its large and widespread use in industry [12]. Therefore, the general population is exposed daily to these substances through oral exposure because of the consumption of contaminated food and water, skin contamination, and intravenous exposure from medical drugs or blood transfusion equipment [13,14]. Numerous studies have investigated the toxic effects of phthalates on health, and it has been established that these compounds can lead to serious adverse effects, such as testicular steroidogenesis disruption, low sperm quality, endometrioses, and infertility [15]. Recent public health studies have established the presence of phthalates and their metabolites in several pregnant women’s physiological samples (placental tissues, cord blood, and amniotic fluid) [16], underlining the fact that the amnion, which is physiologically involved in term rupture, is in direct contact with phthalates and their metabolites throughout pregnancy. Furthermore, it has been established that exposure to these molecules is associated with premature birth [13], a decrease in gestational age [17], miscarriage [18], placental pathologies such as in utero growth restriction (IUGR) [19], and FM pathologies such as pPROM [13]. Because of a physical interaction, several phthalates such as mono(2-ethylhexyl)phthalate (MEHP), which is the active metabolite of DEHP, can act as ligands for intracellular receptors, such as estrogen, androgen receptors, and PPARγ [15], modifying their pathway and action. Recently, it has been established that in the placental area, dysregulation of the PPARγ receptor pathway by MEHP results in an inhibition of trophoblast differentiation, which leads to placental pathologies such as pre-eclampsia and preterm birth [8]. The authors demonstrated that MEHP could interfere with the docking of a specific ligand (rosiglitazone) to PPARγ, leading to a decrease in PPARγ transcriptional activity. Concerning the pathologies of FMs, the link between phthalates, PPARγ, and pPROM has not yet been clearly established; thus, this constitutes the object of the current research project.

In this context, the present study was designed to investigate the effect of MEHP on amniotic cells to determine if this compound can dysregulate the PPARγ receptor pathway, hence resulting in an abnormal level of inflammation molecules, the exacerbation of which could lead to pPROM. After confirming the physiological expression of PPAR in FMs, we studied the impact of MEHP on PPARγ’s expression (mRNA and proteins) and its transcriptional activity, as well as MEHP toxicity and its impact on cell viability and inflammation in a cellular amniotic model (AV3).

## 2. Materials and Methods

### 2.1. Reagents and Antibodies

MEHP (ALR-138N) was purchased from AccuStandard (New Haven, CT, USA). PPARγ agonist rosiglitazone (ab120762), PPARγ antagonist GW9662 (ab 141125), and antibody against PPARγ (ab209350) were purchased from Abcam (Paris, France). For the cell culture, Dulbecco’s modified Eagle medium/nutrient mixture F-12 (DMEM-F12- GlutaMAX) medium and antibiotics (streptomycin, penicillin, and amphotericin B) (CABPS A00-0U), OPTIMEM medium (31985-062), Gibco™ Trypsin-EDTA (0.25%) (25200056) and Lipofectamine 3000 (L3000-015) were obtained from Fisher Scientific (Illkirch-Graffenstaden, France). Moreover, the Superscript IV first-strand-synthesis system, Taq DNA polymerase recombinant (10342020), and Pierce BCA protein assay kit (23225) were also obtained from Fisher Scientific and SYBR Green from Roche (Meylan, France). Charcoal-stripped fetal bovine serum (FBS) (S181F-050), dimethylsulfoxyde (DMSO) (D2650-5), and lipopolysaccharide (LPS) from Escherichia coli O55:B5, which was purified by phenol extraction (L2880-25MG) and used for inflammation tests, were obtained from Sigma (Saint-Quentin-Fallavier, France). The horseradish peroxidase antirabbit conjugated secondary antibody was purchased from Abliance (Compiègne, France).

### 2.2. Cell Culture

The amniocyte epithelial cell line (AV3, CCL-21) was obtained from ATCC. AV3 was cultured under standard conditions (5% CO_2_, 95% humidified air, 37 °C) in DMEM-F12-GlutaMAX supplemented with charcoal-stripped fetal calf serum 10%, antibiotics (ampicillin 100 U/mL and streptomycin 100 µg/mL), and an antifungal (amphotericin B 25 µg/mL).

### 2.3. Tissue Collection and Primary Cell Culture

FMs were collected from nonsmoking women with healthy pregnancies from vaginal or scheduled caesarean deliveries (breech presentation or scarred wombs) (Centre Hospitalier Universitaire Estaing, Clermont-Ferrand, France) after obtaining informed consent. The selected FMs were collected from single pregnant women who had no underlying diseases, no gestational diabetes, and no clinical chorioamnionitis (defined by maternal fever, uterine tenderness, and/or purulent amniotic fluid), as confirmed by the absence of tissue abnormalities after pathologist examination. The FMs were collected throughout all trimesters of pregnancy (first trimester: 1 to 13 weeks of gestation (WG); second trimester: 14–26 WG; third trimester: 27–37 WG; at term: 38–40 WG, by caesarean or vaginal delivery). The research protocol was approved by the institutional regional ethics committee (DC-2008-558). The amnion was dissociated from the choriodecidua. The ZAM (with the thread) and the zone of intact morphology (ZIM, away from the thread) were also distinguished. A suture sewn onto the FMs (from caesarean deliveries) in front of the cervix by the midwife allowed us to identify the ZAM; then, a 4 cm-diameter circle was cut, the ZAM considered, and explant localized places away from the circle boundary were considered in the ZIM. 

The isolation of human primary amniocytes was conducted in three trypsinization steps (10, 20, and 30 min; trypsin-EDTA 0.25%) at 37 °C, followed by scraping of the amnion. Cells were filtered to remove the collagen, centrifuged for 5 min at 1000 rpm, and grown in T75 flasks coated with collagen type I (04902; Stem Cell Technologies, Vancouver, BC, Canada). The amniocytes were cultivated under standard conditions (5% CO_2_, 95% humidified air, 37 °C) in complete Dulbecco modified Eagle medium F-12 nutrient mixture supplemented with 10% fetal bovine serum, 100 μg/mL streptomycin, 100 IU/mL penicillin, and 250 μg/mL amphotericin B.

### 2.4. Characterization of PPAR Isoform by RT-PCR on FM Explants and Cells

After the disruption step with a Precellys homogenizer (Bertin Technologies, Montigny-le-Bretonneux, France) using ceramic beads (KT03961, Ozyme, Saint-Cyr-l’École, France), the total RNA was extracted from the FMs explants using RNAzol^®^ RT (RN190, Molecular Research Center, Cincinnati, OH, USA). For the cells, RNA extraction was performed with NucleoSpin RNA (740955.50, Macherey-Nagel, Hoerdt, France), following the manufacturer’s instructions. Retrotranscription of RNA extracted in cDNA was made from 1 µg of total RNA using the SuperScript IV First-Strand Synthesis System for reverse transcription polymerase chain reaction (RT-PCR), according to the manufacturer’s protocol. After this step, polymerase chain reaction (PCR) experiments were performed using oligonucleotides for PPARα, PPARβ, and PPARγ (Table 1). The results were analyzed on a 2% agarose gel. The housekeeping gene RPL0 (36b4, Acidic Ribosomal Phosphoprotein P0) was chosen as a witness to the presence of cDNA and the proper functioning of the PCR reaction.

### 2.5. Treatments

#### 2.5.1. MEHP Treatments

AV3 cells were seeded in six-well plates at a quantity of 3 × 10^5^ cells/well for all experiments, except for cell viability, where the cells were incubated at a rate of 10^4^ cells/well in 96-well plates. The next day, the cells were pretreated with 0.2% DMSO or MEHP at 1, 10, or 100 µM or a PPARγ antagonist GW9662 at 1 µM. After 3 h, PPARγ’s agonist rosiglitazone (RGT) at 1 µM was added, and the cells were incubated (5% CO_2_, 95% humidified air, 37 °C) for 24 and 48 h. DMSO was used as a negative control because of its role in diluting solvents for MEHP and the PPARγ agonist and antagonist. These treatments are used to determine MEHP’s impact on cytotoxicity, cell viability, and PPARγ expression (RNA and proteins) and activity (luciferase assay). The results are expressed as the fold change between the treatment conditions and control group (DMSO).

#### 2.5.2. Inflammatory Treatment with LPS

AV3 cells were seeded in six-well plates at a quantity of 3 × 10^5^ cells/well. The next day, the cells were pretreated for 3 h with MEHP at 1, 10, or 100 µM or PPARγ antagonist GW9662 at 1 µM. After 3 h, the cells were treated with rosiglitazone at 1 µM for 12 h. Then, the medium was removed and replaced with a fresh one. The cells were pretreated again with MEHP at 1, 10, or 100 µM or antagonist for 3 h. Then, 1 µM of the PPARγ agonist was added to LPS (10 µg/mL), and the cells were incubated for 24 and 48 h at 37 °C with 5% CO_2_. This treatment was used to determine the impact of MEHP on the inflammatory process. RNA and protein expressions of IL-6 and IL-8 were measured. 

### 2.6. Cytotoxicity

For evaluating the impact of the phthalates’ treatment on cell damage, the release of the intracellular enzyme lactate dehydrogenase (LDH) into the cell media was measured. The activity of LDH was quantified on a machine automate (Vista, Siemens Health Diagnosis, Paris, France) using an enzymatic assay, following the manufacturer’s recommendations. The results are expressed as the fold change between treatment conditions and the control group (DMSO).

### 2.7. Cell Viability

MEHP impact on cell viability was analyzed using the Cell-counting Kit-8 assay (ab228554, Abcam, Paris, France) at 24 and 48 h post treatment, using an enzymatic assay and following the manufacturer’s recommendations. The results are expressed as the fold change between treatment conditions and the control group (DMSO).

### 2.8. Quantitative RT-PCR 

RNA and cDNA were obtained in the same way as previously described for PPAR isoform characterization. PPARγ, IL-6, and IL-8 expressions were assessed by quantitative RT-PCR (RT-qPCR) performed using LightCycler^®^ 480 SYBR Green I Master (Roche, Meylan, France) with specific oligonucleotides (Table 2). Transcript quantification was performed. Standard curves were used to quantify the amount of amplified transcript. The results were normalized to the geometric mean of the human housekeeping genes RPL0 (36b4) and RPS17 (acidic ribosomal phosphoprotein P0 and ribosomal protein S17, respectively), as recommended by the MIQE guidelines [20].

### 2.9. Western Blot Analysis

The extraction of total proteins from FM explants (after grinding) and cells was carried out in agreement with the protocol of the “Membrane Protein Extraction” kit (BioVision). The protein concentration was determined using a BCA protein assay kit. The samples were denatured in a loading buffer containing 3% β-mercaptoethanol at 100 °C for 5 min and separated (40 µg of total protein per well) on 4% to 15% Mini-PROTEAN TGX Stain Free Gels (4568084, BIO-RAD, Marnes-la-Coquette, France). After electrophoresis, proteins were transferred onto nitrocellulose membranes using a semidry transfer cell; this was performed using Trans Blot Turbo RTA Midi Transfer Kit Nitrocellulose (1704150, BIO-RAD, Marnes-la-Coquette, France). Nonspecific sites were blocked for 45 min with Tris-buffered saline (TBS) containing 5% nonfat dry milk (blocking solution) and then incubated overnight at 4 °C with PPARγ antibody (1/1000). The membrane, which was rinsed three times with TBS 0.1% Tween 20, was incubated for 1.5 h at room temperature with a horseradish peroxidase antirabbit conjugated secondary antibody (1/10,000) in a hybridization solution. After three wash cycles in TBS-0.1% Tween 20, the membrane peroxidase activity was assayed by enhanced chemiluminescence (Clarity Western ECL Substrate, BIO-RAD, Marnes-la-Coquette, France). The relative intensities of the protein bands were analyzed using Image Lab software (BIO-RAD, Marnes-la-Coquette, France) (Appendix A). The results are presented as a ratio between the protein of interest and total protein on the same blot.

### 2.10. PPARγ Gene Reporter Luciferase Assay

AV3 cells were seeded in six-well plates at a concentration of 3 × 10^5^ cells/well. Then, the cells were transfected with two plasmids using Lipofectamine 3000. The first vector was pGL3-J3-TK-Luc, which contains the luciferase reporter gene the transcription of which depends on three PPRE fixation sites, and the other vector was PCH110 (vector kindly gifted by the laboratory of Prof. Pierre CHAMBON (IGBMC, Illkirch-Grafenstaden, France)) containing the gene LAC-Z and encoding β-galactosidase. The amount of each vector was 0.5 µg pGL3-J3-TK-Luc and 0.15 µg of PCH110. After 4 h of transfection in OPTIMEM medium, the transfection media were replaced by a medium containing the treatments (see treatments paragraph). After 24 h incubation at 37 °C under 5% CO_2_, the cells were packed and kept at −20 °C.

Measurement of luciferase activity: The cells were lysed in 120 µL of 1X lysis buffer from the Luciferase Reporter Gene Assay kit (11814036001, Roche, Meylan, France). After 5 min of incubation at room temperature, the cellular debris were eliminated by centrifugation (3 min, 13,000 rpm), and supernatant (20 µL) was mixed with 50 µL buffer containing the enzyme substrate. Under the action of luciferase, the substrate was transformed into a luminescent product for which the measured intensity—with a FB12 luminometer (Berthold, Thoiry, France)—was proportional to the activity of luciferase, which itself depends on PPARγ receptor activity.

Measurement of β-galactosidase activity: In a 96-well plate, 50 µL of supernatant was incubated for 10 min at 37 °C in the dark with 50 µL of the reagent from the kit “Mammalian, β-galactosidase Assay kit” (75707, ThermoScientific, Illkirch-Graffenstaden, France) containing the enzyme substrate. Βeta-galactosidase transforms this substrate into a yellow product color. Its optical density, measured at 405 nm using a Multiskan^®^ spectrophotometer Spectrum (Thermo Scientific, Illkirch-Graffenstaden, France), is proportional to the activity of the enzyme. The results were then obtained by carrying out the Luciferase/β-galactosidase report. The final results are expressed as the fold change between the treatment and control group (DMSO 0.2%).

### 2.11. Cytokine Release Assay

The release of IL6 and IL8 in the culture media was quantified after 48 h of treatment (see the inflammatory treatments paragraph) using automated multiplex immunoassays on ELLA™ (San Jose, CA, USA). The total protein concentration was determined using a BCA protein assay kit. Cytokine concentrations were normalized to total protein concentration, and the fold change “treated/LPS” was reported.

### 2.12. Statistical Analysis

The data are expressed as the mean ± standard error of the mean and are an average of the duplicates or triplicates of at least four independent experiments. Because the results did not follow a normal distribution, a comparison of means was performed using nonparametric tests. First, a one-way ANOVA Kruskal–Wallis was realized to study a global comparison between all the groups, followed by multiple comparison with Dunn’s correction for comparing more than two groups, using PRISM software 5.02 (GraphPad Software Inc., San Diego, CA, USA). For all studies, the values were considered significantly different at *p* < 0.05 (*), *p* < 0.01 (**), and *p* < 0.001 (***).

## 3. Results

### 3.1. Characterization of PPAR Isoforms (α, β, γ) Expression (mRNA, Protein) within FMs and Amniotic Cells

The mRNA expression profiles of the three isoforms of PPAR (α, β, γ) in FMs were investigated on amnion and choriodecidua samples throughout pregnancy trimesters. The RT-PCR experiments revealed that FMs expressed the three isoforms of PPAR in both amnion and choriodecidua during each trimester, regardless of the delivery mode (caesarean or vaginal) and morphologic zone (ZIM and ZAM). The PPAR transcripts were also expressed in primary amniocytes (Amn I) and the AV3 cell line (Figure 1A). PPARγ protein expression was also demonstrated in primary amniocytes, the AV3 cell line, and both layers of FMs obtained after vaginal delivery or caesarean section, here with a separate consideration of the ZIM and ZAM (Figure 1B). PPARγ expression was compared by RT-qPCR in the two FM layers at two different times of the pregnancy, considering only the second and third trimester because the two layers were not dissociated at the first trimester. RNA expression of PPARγ was significantly decreased at the third trimester in the amnion compared with the second trimester, and PPARγ was less expressed in amnion compared with chorion either in ZIM or ZAM (Figure 1C).

### 3.2. Effect of MEHP on Cell Viability and Cytotoxicity

Because of the ubiquitous presence of phthalates in the environment, the absence of the phthalate contamination of consumables and reagents was validated. The concentration of DEHP, which is the precursor of MEHP, was measured by gas chromatography coupled with a mass spectrometer (GC/MS) in the culture mediums before and after incubation in plastic consumables (data not shown). These concentrations were analyzed three times for each sample and were below the limit of detection (LOD) of 0.03 µg/mL, regardless of the condition tested. We concluded that the only exposure to phthalates resulted from treatment with these compounds.

To study the effects of MEHP on cell viability and cytotoxicity, AV3 cells were exposed at different concentrations of MEHP (1, 10, or 100 µM) for 24 or 48 h. The DMSO results are similar to those obtained with the non-treated condition, composed of media alone. The results demonstrate that was no effect of MEHP on cell viability and cytotoxicity, regardless of the dose and time of treatment (Figure 2). In addition, similar results were observed after treatments with MEHP, GW9662, or rosiglitazone alone, or rosiglitazone combined with GW9662 or MEHP.

### 3.3. Effects of MEHP on PPARγ Expression and Transcriptional Activity in AV3 Cells

To assess the involvement of PPARγ in the response elicited by MEHP, the effects of MEHP on the mRNA and protein expression of PPARγ in the AV3 cell line were studied by RT-qPCR and Western blot analysis. Here, MEHP, GW9662, or rosiglitazone single exposure (for 24 and 48 h) did not alter the levels of the PPARγ transcripts and proteins. The same results were observed with the combined treatment (rosiglitazone with GW9662 or MEHP) (Figure 3 and Appendix A).

To investigate the effect of MEHP on PPARγ transcriptional activity, the luciferase gene reporter assay was used. First, to validate this test, we treated AV3 cells by rosiglitazone and/or GW9662 at 1 µM (Figure 4). As expected, rosiglitazone induced an increase in PPARγ (*p* = 0.0099) transcriptional activity, and this was significantly inhibited by GW9662 (*p* = 0.0286). MEHP treatments alone, regardless of the dose used, had no effect on luciferase expression, and combined treatment with MEHP and rosiglitazone (Figure 4) suppressed the transcriptional activity induced by this agonist (*p* = 0.029 for MEHP 1 µM, *p* = 0.032 for MEHP 10 µM, and *p* = 0.016 for MEHP 100 µM). 

### 3.4. Effect of MEHP on Anti-Inflammatory Properties of PPARγ

Considering that MEHP modulates the transcriptional activity of PPARγ and the well-known anti-inflammatory properties of this nuclear receptor, the effects of MEHP on the inflammatory process were investigated. AV3 cells were incubated with LPS to induce an inflammatory status (1,5-fold induction), and different combinations of treatment were tested. The RNA and protein expressions of well-known inflammatory cytokines (IL-6 and IL-8) were analyzed by RT-qPCR (Figure 5A) and ELLA assay (Figure 5B) after 24 and 48 h of treatment, respectively. Treatment by rosiglitazone for 24 h decreased RNA expressions of IL-6 (*p* = 0.0125) and IL-8 (*p* = 0.0073) in LPS-treated FMs, confirming the anti-inflammatory properties of PPARγ, whereas treatment with GW9662 alone (Figure 5A) or MEHP alone (data not shown) in these membranes had no direct effect on cytokine expression. MEHP at 100 µM removed an agonist-induced decrease for both cytokines studied, and the concentration of MEHP at 10 µM only decreased IL-8 expression. When the AV3 cells were treated with rosiglitazone for 48 h, the protein expression of IL-6 (*p* = 0.0294) and IL-8 (*p* = 0.0081) decreased significantly (Figure 5B). Interleukin protein expression was also studied with cotreatment with rosiglitazone and MEHP. For Il-6, MEHP at 100 µM counteracted the agonist-induced decrease. Similar results were obtained for IL-8 protein expression with MEHP at 10 µM and 100 µM.

## 4. Discussion

Although phthalates are known to be reprotoxic and their presence has been identified in the urine of pregnant women, few studies have evaluated the effects of these compounds in pregnant women. Nevertheless, it has been shown that phthalates can cross the placental barrier and FMs, because their presence has been observed in placental tissues, the umbilical cord, and amniotic fluid [21,22,23]. However, some recent works have observed a link between phthalate exposure and the occurrence of obstetrical pathologies, whether of placental origin, such as IUGR and pre-eclampsia, or because of the abnormal fragility of the FM [16], such as pPROM. One proposed molecular explanation of the obstetrical alteration by phthalates is their ability to alter the signaling pathway supported by the nuclear receptor PPARγ. If the link between phthalates and this receptor has been relatively well studied in placenta [8,18], no study has focused on the deregulation of the activity of this receptor in FMs during pPROM. Moreover, considering the essential character of PPARγ during pregnancy and its anti-inflammatory action in FMs, it seemed interesting to study the impact of the toxic compounds in fetal membranes and to determine if such chemicals can be a factor in the preterm rupture of these membranes. 

Our study established that PPARγ is expressed during the three trimesters in FMs, leading to early and continuous potential interactions with the phthalates during pregnancy. This is more evident for the amnion in direct contact with the amniotic fluid, where the phthalates accumulate throughout pregnancy; hence, this layer is more likely to suffer from the phthalates’ toxic properties. In addition, the amnion expresses PPARγ at lower levels than the choriodecidua and could present decreased PPARγ molecular actions as a result. With a lower expression of PPARγ in the third trimester, the amnion will be less protected against inflammation by this nuclear receptor at the end of pregnancy. The phthalates could then interfere with this specific process, which could be exacerbated and take place in the global dysregulation of molecular events (oxidative stress, ECM degradation, senescence, etc.) leading to earlier and more intense weakening of the FM. 

To establish this PPARγ dysregulation, we used one the most present phthalates in amniotic fluid, that is, MEHP, and the amniotic cell line AV3, here modeling the direct contact of epithelial cells of the inner fetal membrane with the amniotic fluid. This interference was established using three doses of MEHP (1, 10, and 100 μM), covering the concentration ranges of this compound observed in the physiological fluids of pregnant women [21,22,23]. Regardless of the doses tested, MEHP was not toxic for amniotic cells and did not affect their viability. Therefore, MEHP did not enhance cellular death, which is one of the cellular mechanisms weakening FMs at the end of pregnancy. We also did not observe a decrease in terms of PPARγ expression, excluding this molecular mechanism in explaining the dysregulation occurring from phthalates on this nuclear signaling pathway. Several recent research works have demonstrated that PPARγ expression (mRNA and protein) can be altered with different doses of MEHP in the placenta area. However, mixed results depending on the cell lines used have been found, thus demonstrating that MEHP’s impact on PPARγ expression is cell and tissue specific [8,24]. By contrast, treatment for 24 h with a PPARγ agonist (rosiglitazone) induced a significant increase in nuclear receptor activity, whereas the use of an antagonist (GW9662) abolished this induction, confirming the functionality of the current study’s model. MEHP alone does not modify the activity of the receptor activity, indicating that this compound does not behave as an agonist of the receptor. On the other hand, these experiments have demonstrated that MEHP suppressed the agonist-induced activation of the PPARγ receptor, thus testifying to the antagonistic effect of this phthalate. This effect could be explained in particular by the fact that MEHP decreases the capacity of PPARγ to release its coregulator NCoR, but also its ability to recruit coactivators (p300), as demonstrated during crystallographic studies [25]. This selective recruitment of coregulators results in a partial or null transcription of PPARγ-regulated genes and, thus, a partial activity of the receptor.

The receptor PPARγ is a nuclear receptor, which has been described as a potential target to fight against inflammation [9,10]. Several studies show that when one (thiazolidinedione) of its ligands activates this receptor, the expression of proinflammatory protein (e.g., IL-6 or IL-8) was decreased [26]. In the present study, it was demonstrated that the MEHP, by deregulating PPARγ pathway activation, could deregulate its anti-inflammatory properties in amniotic cells. In fact, in a proinflammatory environment (created with LPS), the expression of IL-6 and IL-8 was increased when the cells, and hence the receptors, were in the presence of a combination of MEHP and agonist. Thus, MEHP, in the FMs context, can emphasize the inflammation already present, which can be linked to a premature weakening of the membrane and can induce pPROM. Moreover, this receptor is also known to be able to control the gene involved in oxidative stress regulation [27], which is another process that can be linked to FM weaknesses. It has already been demonstrated that MEHP is able to increase oxidative stress [28] in FMs, so it will be interesting to study if, through the PPARγ signaling pathway, this compound could also deregulate this molecular event involved in FM rupture.

In this study, a cell line model (AV3) was used to assess the impact of MEHP on the PPARγ amniotic pathway. Even if it was originally obtained using epithelial amniotic cells, the ATCC supplier indicated that this cell line was contaminated by HELA cells. Nevertheless, this cell line has been characterized for use as a model for the investigation of prostaglandin–cytokine interactions in human amnion. The AV3 cell line presents the same basal levels of IL-6 and IL-8 as those observed in primary amnion cells, which is why this cell line was used in this study [29]. In addition, we demonstrated that the AV3 cell line naturally expressed PPARγ, excluding the transfection of PPARγ plasmid expression, leading to a more physiological model of amniotic epithelial cells. The first results need to be confirmed with the use of primary amniotic cells.

In conclusion, by negatively interacting with the amniotic PPARγ signaling pathway, the phthalates could exacerbate the inflammation, one of the processes occurring during the physiological rupture of FM. This exacerbation could lead to the pathological weakening of FM found in 30–40% of preterm births. 

## Figures and Tables

**Figure 1 life-12-00544-f001:**
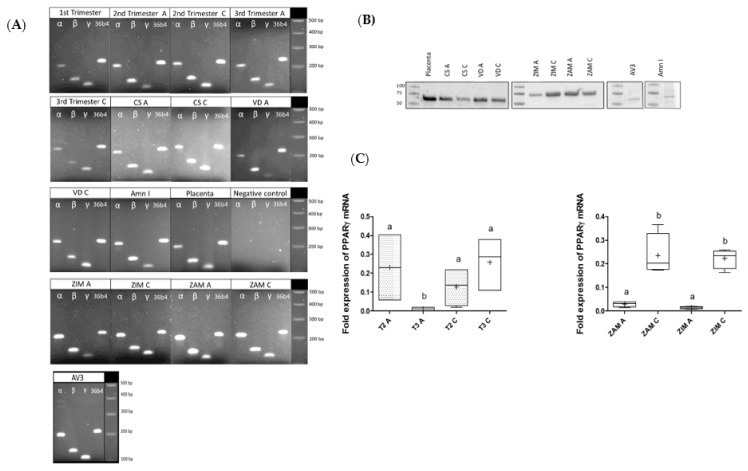
Characterization of PPAR isoform expression (α, β, γ) within FMs and related cells. (**A**) Three PPAR isoform (α, β, γ) transcripts were checked by RT-PCR assays in the two layers of fetal membranes (FMs), amnion (A) and choriodecidua (C), throughout pregnancy (three trimesters), whatever the mode of delivery (caesarean (CS) or vaginal delivery (VD)) and whatever the morphological zone (ZIM and ZAM). The receptor transcripts were also checked in primary amniocytes isolated from FMs. The placenta sample was used as a positive control for PPAR expression. (**B**) PPARγ proteins were checked by Western blot assays on amnion (A) and choriodecidua (C) samples from term and third trimester FMs (caesarean (CS), vaginal delivery (VD), ZIM and ZAM, primary amniocytes isolated from FMs, and AV3 cell line). The placenta sample was used as a positive control for PPARγ expression. (**C**) Transcripts of PPARγ were measured by RT-qPCR in amnion (A) and in the choriodecidua (C) either in the ZIM or ZAM at the second (T2) and third trimester (T3) after cesarian delivery. The results were normalized to the geometric mean of the human housekeeping genes RPL0 and RPS17, as recommended by the MIQE guidelines. A comparison of conditions was realized by a Kruskal–Wallis one-way ANOVA test, followed by a Dunn’s post hoc test: a ≠ b, *p* < 0.05. Results are presented in Tukey boxes, and means are indicated by “+”.

**Figure 2 life-12-00544-f002:**
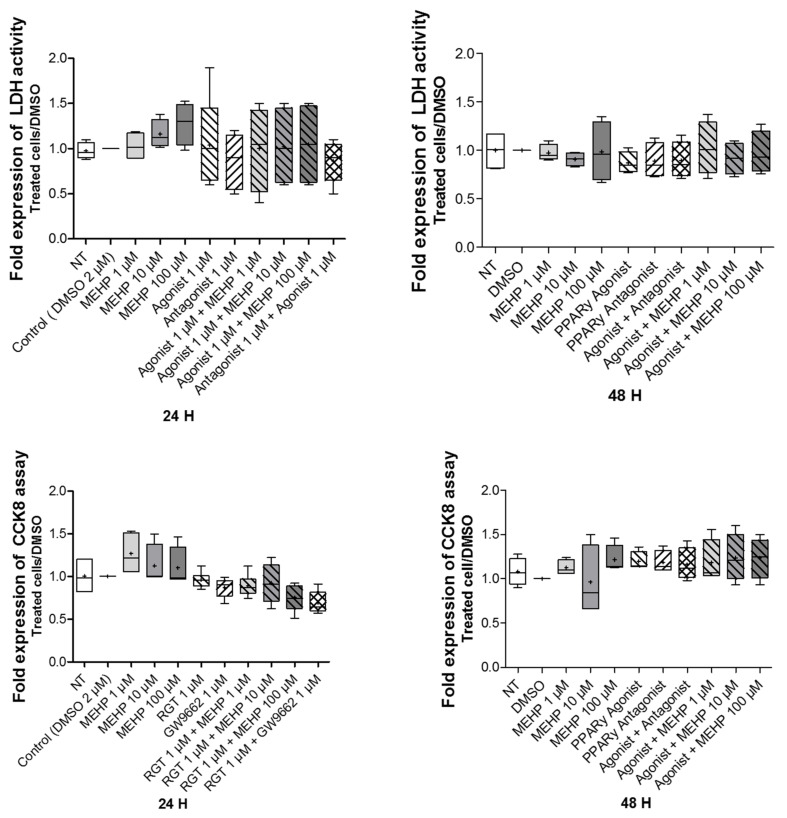
Impact of MEHP on AV3 cell viability and toxicity. Upper panel: MEHP toxicity was evaluated by LDH release measurement in a culture medium. Lower panel: cell viability was studied by a Cell-counting Kit 8 (CCK8) assay (lower panel). AV3 cells were treated for 24 or 48 h with DMSO and MEHP at 1, 10, or 100 µM, or PPARγ antagonist (GW9662) at 1µM and/or PPARγ agonist rosiglitazone (RGT) at 1 µM. For each condition tested, the results were reported against DMSO (negative control). Each condition was performed in duplicate, and the experiment was repeated four times. The results were analyzed by ANOVA statistical test (Kruskal–Wallis test) followed by Dunn’s post hoc test. Results are presented in Tukey boxes, and means are indicated by “+”.

**Figure 3 life-12-00544-f003:**
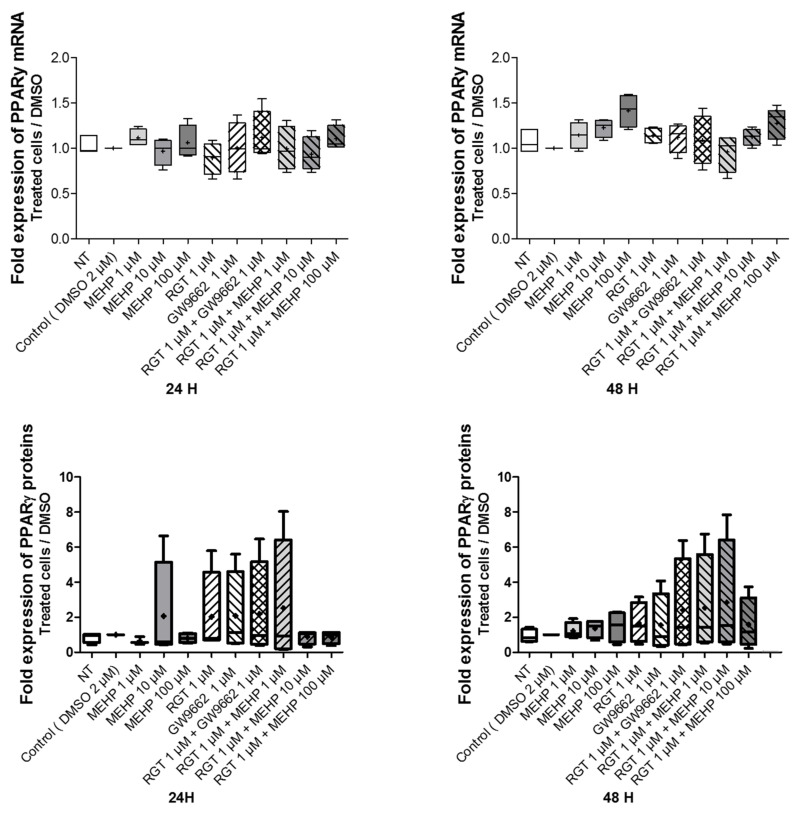
Impact of MEHP on PPARγ mRNA and protein expression. Upper panel: Expression of PPARγ transcripts as evaluated by quantitative RT-qPCR. Lower panel: PPARγ protein expression as evaluated by Western blot assay. AV3 cells were treated for 24 and 48 h with DMSO and MEHP at 1, 10, or 100 µM or PPARγ antagonist (GW9662) at 1 µM and/or PPARγ agonist rosiglitazone (RGT) at 1 µM. Each condition was performed in duplicate, and the experiment was repeated five times. The results were analyzed by an ANOVA statistical test (Kruskal–Wallis test) followed by Dunn’s post hoc test and are expressed as a fold change between the treated conditions compared with DMSO. Results are presented in Tukey boxes, and means are indicated by “+”.

**Figure 4 life-12-00544-f004:**
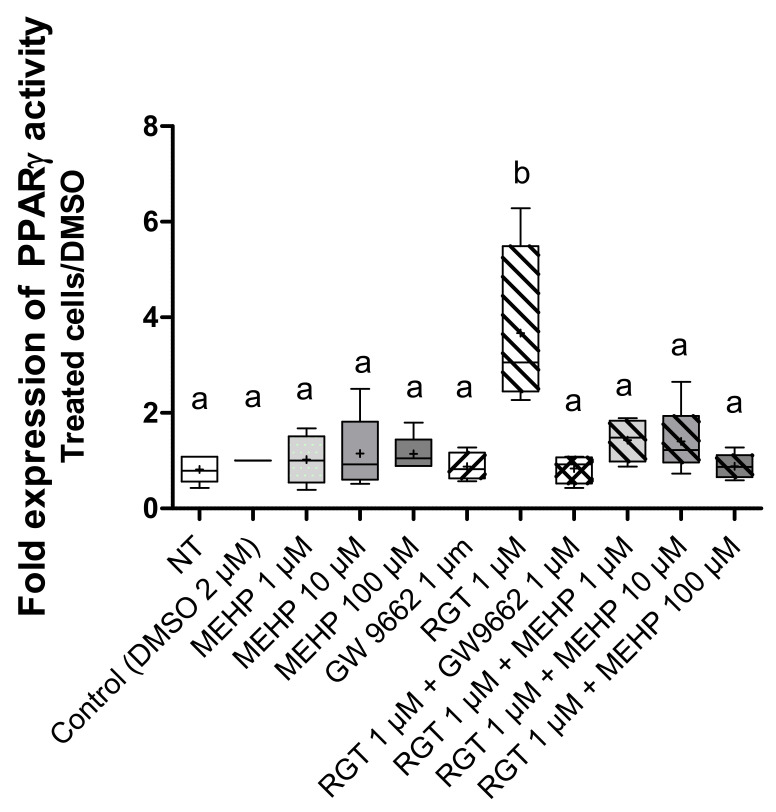
Effect of MEHP on PPARγ transcriptional activity. PPARγ transcriptional activity was assayed by a bioluminescence reaction using the “Luciferase Reporter Gene Assay” kit (Roche Applied Science). AV3 cells were treated for 24 h with DMSO and MEHP at 1, 10, or 100 µM, or PPARγ antagonist (GW9662) at 1 µM and/or PPARγ agonist rosiglitazone (RGT) at 1 µM. Each experiment was performed in duplicate, and the experiment was repeated four times. The results are expressed as a fold change between the treated conditions and DMSO and were analyzed by ANOVA statistical test (Kruskal–Wallis test) followed by Dunn’s post hoc test: a ≠ b, *p*
*<* 0.01. Results are presented in Tukey boxes, and means are indicated by “+”.

**Figure 5 life-12-00544-f005:**
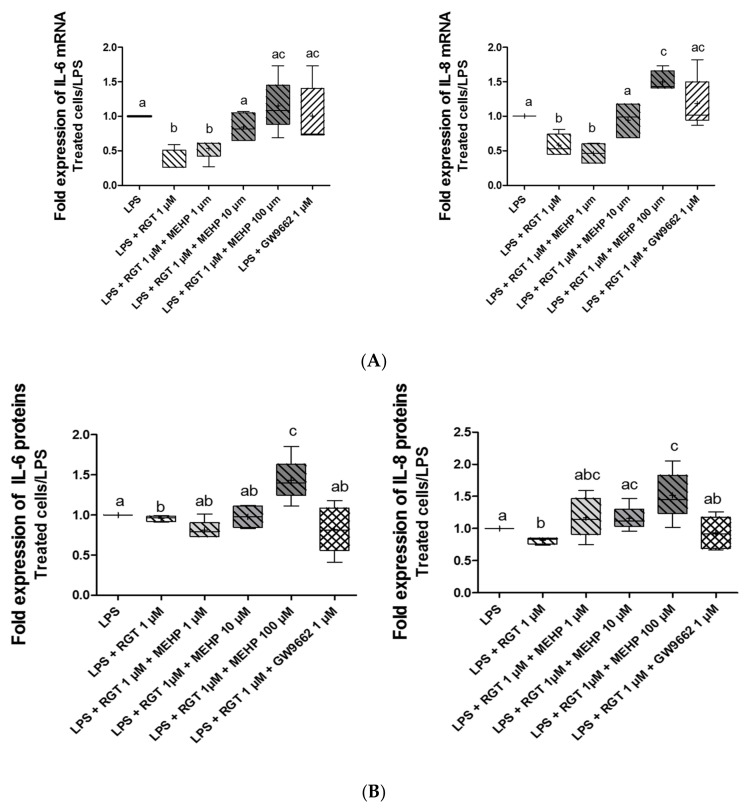
Effect of MEHP on the anti-inflammatory properties of PPARγ. RNA and protein expression of cytokines (IL-6 and IL-8) were analyzed by RT-qPCR (**A**) and ELLA assay (**B**) after 24 (upper panel) and 48 h (lower panel) of treatment, respectively. AV3 cells were treated with LPS (10 µg/mL) and MEHP at 1, 10, or 100 µM, or a PPARγ antagonist (GW9662) at 1µM and/or PPARγ agonist rosiglitazone (RGT) at 1 µM. Each condition was performed in duplicate, and the experiment was repeated four times. The results were analyzed by an ANOVA statistical test (Kruskal–Wallis test) followed by Dunn’s post hoc test: a ≠ b ≠ c, *p*
*<* 0.05. Results are presented in Tukey boxes, and means are indicated by “+”.

**Table 1 life-12-00544-t001:** Forward and reverse primer sequences used for RT-PCR amplification of human genes.

Gene	Sequence 5′-3′ (F: Forward, r: Reverse)	Product Size (bp)	Annealing Temperature (°C)
hsPPARα	F: GATCTAGAGAGCCCGTTATC	208	61
	R: GGACCACAGGATAAGTCAC		
hsPPARβ	F: AGTGCCTGGCACTGGGCATG	222	61
	R: TCAGGTAGGCATTGTAGATGTGC		
hsPPARγ	F: AGTGGGGATGTCTCATAATGCC	117	61
	R: GCAGAGTTTCCTCTGTGATA		
hs36b4	F: AGGCTTTAGGTATCACCACT	219	61
	R:GCAGAGTTTCCTCTGTGATA		

**Table 2 life-12-00544-t002:** Forward and reverse primer sequences used for RT-qPCR amplification of human genes.

Gene	Sequence 5′-3′ (F: Forward, r: Reverse)	Product Size (bp)	Annealing Temperature (°C)
hsIL6	F: AATGAGGAGACTTGCCTGGTG	143	61
	R: AGGAACTGGATCAGGACTTTTG		
hsIL8	F: TGATTTCTGCAGCTCTGTGTG	154	61
	R: TCTGTGTTGGCGCAGTGTGG		
hs36B4	F: AGGCTTTAGGTATCACCACT	219	61
	R: GCAGAGTTTCCTCTGTGATA		
hsRSP17	F: TGCGAGGAGATCGCCATTATC	169	61
	R: AAGGCTGAGACCTCAGGAAC		

## Data Availability

The data presented in this study are available on request from the corresponding author. Western Blot data presented in this study are available Appendix A.

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
