# Peer review of "Dysregulation of the Amniotic PPARγ Pathway by Phthalates: Modulation of the Anti-Inflammatory Activity of PPARγ in Human Fetal Membranes"

_life, 2022, doi:10.3390/life12040544_

Round 1
Reviewer 1 Report
The manuscript entitled "Dysregulation of the amniotic PPARγ pathway by phthalates: A new hypothesis to explain the premature rupture of fetal membranes" is intriguing yet requires adding data to prove the hypothesis.
The fetal membranes were collected from pregnant women in different trimesters of pregnancy. There is no data about the levels of phthalates (MEHP) in these pregnant women with healthy pregnancies.
This manuscript does not have transfection data, although these are mentioned in the methodology. The rationale to hypothesize that MEHP can selectively target nuclear transcription factor PPARg is not established with present data. It is also unclear how phthalate is selectively upregulating PPARg in the amniotic membrane.
The problems in describing statistical comparisons among exps. groups are critical.
LPS activated FM and MEHP induced FM does not produce a similar cellular response. The null functional changes during PPAR silencing could probably differentiate these two.
MEHP metabolites can act as ligands for the endocrine receptor; however, those do not specify effects on PPARg selectively. The other important issue has that amniotic membrane/amnion and amniotic cell line (AV) essentially differed. AV3 is derived via HeLa contaminated adenocarcinoma.
The assumption that abnormal inflammation leads to PROM does not depend on the PPARg dysregulation pathway. Therefore, causality and association should be dealt with using discrete data.
Data are not presented properly, both structurally and functionally.
The bar/column diagram should be labelled individually within each figure. Several figures showed no significant changes, yet it is placed (For example, Fig.2 and others). These figures can be skipped, and data should be placed within the text.
In figure 1. B. expression of PPAR-g protein was checked by western blot. Housekeeping protein used for normalization is not mentioned, and data regarding the same is not presented in this manuscript.
Fig.2 -DMSO 2uM is used as a negative control in a proliferation assay. However, there are no media controls used.
The western blot performed (Fig. 3) lower panel PPAR-g protein in AV3 cells. The protein blots are missing.
Fetal membrane weakening is induced by the combination of events such as oxidative stress, apoptosis, senescence, extracellular matrix degradation. Data on oxidative stress and ECM degradation in the presence of phthalate could have strengthened the manuscript.
Effects of MEHP exposure on PPAR-g target genes could be helpful.
The discussion requires a revision with a specific explanation of their data. The statement on phthalate and PPARg is largely exaggerated. PPARg expression is an agonist response via several upstream mediators of the external stimuli like this compound (MEHP). Lack of phenotypic data, rationale, data presentation makes the conclusion weak.
Author Response
Comments from Reviewer 1
The authors thank the reviewer 1 for the constructive comments and for the help to raise the paper level.
~ Comment 1: The fetal membranes were collected from pregnant women in different trimesters of pregnancy. There is no data about the levels of phthalates (MEHP) in these pregnant women with healthy pregnancies.
Answer 1: First, thank reviewer 1 for pointing this out. Indeed, the level of phthalates in the fetal membranes collected from healthy pregnancies was not included in the design of the study. Nevertheless, these membranes were only used to evaluate the PPAR expression level during the pregnancy and were not used to determine PPAR pathway modulation by phthalates. As the authors demonstrated that MEHP do not modulate PPARg transcripts and proteins expression, the determination of MEHP levels in these membranes is considered as not necessary.
~ Comment 2: This manuscript does not have transfection data, although these are mentioned in the methodology. The rationale to hypothesize that MEHP can selectively target nuclear transcription factor PPARg is not established with present data. It is also unclear how phthalate is selectively upregulating PPARg in the amniotic membrane.
Answer 2: The authors well performed transfection experiments using lipofectamine 3000 as described in the paragraph 2.10: PPARγ gene reporter luciferase assay. As indicated, they also demonstrated that MEHP selectively target nuclear transcription pathway (paragraph 3.3). Indeed, as presented in this paragraph, a well-established agonist of PPARg (rosiglitazone 1 µM) induces an increase of transcriptional activity measured by a production of luminescence (luciferase reporter gene assay with a specific PPAR responsive element). This increase is selectively inhibited by MEHP as represented in Figure 4. Moreover, PPARg transcripts and proteins are not direct target of MEHP because there is any change in term of expression after MEHP treatment (Figure 3). All these results demonstrated a direct dysregulation of PPARg by MEHP, in terms of transcriptional activity.
~ Comment 3: The problems in describing statistical comparisons among exps. groups are critical.
Answer 3: The authors agreed with this comment and have modified the text of paragraph 2.12 (line 40 to 44; page 6) to clarify the statistical tests. In order to apply this change, the legend of the figure 2, 3, 4 and 5 were also modified to indicate that a comparison of different conditions was realized by Kruskal-Wallis one-way ANOVA test followed by Dunn’s post-test.
~ Comment 4: LPS activated FM and MEHP induced FM does not produce a similar cellular response. The null functional changes during PPAR silencing could probably differentiate these two.
Answer 4: To clarify this important point, the MEHP presents a pro-inflammatory action by inhibiting the anti-inflammatory action of PPARg agonist. Alone MEHP was not able to support an intrinsic pro-inflammatory property. In this context, the authors modified the first version of paragraph 3.4 (line 1 to 5; page 12) follow: “Treatment by rosiglitazone for 24 h decreased RNA expressions of IL-6 (p=0.0125) and IL-8 (p=0.0073) in LPS treated FMs, confirming the anti-inflammatory properties of PPARγ, whereas treatment with GW9662 alone (Figure 5 A) or MEHP alone (data not shown) in these membranes had no direct effect on cytokine expression”.
~ Comment 5a: MEHP metabolites can act as ligands for the endocrine receptor; however, those do not specify effects on PPARg selectively.
Answer 5a: The authors agree with the reviewer that MEHP metabolites could act as a ligand for endocrine receptor as previously published for glucocorticoid receptor or estrogen receptor. But the authors demonstrated that MEHP inhibits the anti-inflammatory properties of a specific agonist of PPARg i.e. rosiglitazone. As this ligand was unable to be a ligand of the other nuclear receptor, that demonstrated the PPARg selectively of phthalate in the inflammatory context.
~ Comment 5b: The other important issue has that amniotic membrane/amnion and amniotic cell line (AV) essentially differed. AV3 is derived via HeLa contaminated adenocarcinoma.
Answer 5b: The authors agree with the reviewer’s information about the AV3 cell line. Nevertheless, this cell line has been derived from amniotic cell as stipulated by ATCC catalogue. This cell line is usually used in published papers dealing with amniotic context. For example the authors’ team recently published a paper to study the physiological TLR4 regulation in human FM as an explicative mechanism of pathological preterm case (Belville et. al, e-Life 2022). Moreover, another interesting characteristic of this cell line is that it express naturally PPARg, excluding transfection of PPARg plasmid expression.
~ Comment 6: The assumption that abnormal inflammation leads to PROM does not depend on the PPARg dysregulation pathway. Therefore, causality and association should be dealt with using discrete data.
Answer 6: Inflammation is an event which has been clearly observed in case of PPROM as Menon et al. confirm in their review about “Fetal membrane architecture, aging and inflammation in pregnancy and parturition” (oi: 10.1016/j.placenta.2018.11.003. ). The authors are confident about their results, which demonstrated that phthalates disrupt the anti-inflammatory properties of PPARγ in amniotic area. In this context, phthalate could be consider as pro-inflammatory agent, which could be a molecular explanation of fetal membranes weakening leading to PPROM. Nevertheless, the authors agreed with the general comment recommending using a more tempered language for PPROM hypothesis and moderate the tone of their text as for example, proposing a new title of the revised manuscript “ Dysregulation of the amniotic PPARg pathway by phthalates: modulation of the anti-inflammatory activity in human fetal membranes.
~ Comment 7: Data are not presented properly, both structurally and functionally. The bar/column diagram should be labelled individually within each figure. Several figures showed no significant changes, yet it is placed (For example, Fig.2 and others). These figures can be skipped, and data should be placed within the text.
Answer 7: The authors did not understand the first part of the comment. All the figures/ bar-column diagrams were already labeled individually in the first version of the manuscript. Usually, the authors agree with the proposal of the reviewer to skip figure without significant changes. In the case of this study, it seems slightly out of scope because one indirect explanation of PPARg pathway disruption could be the modification of PPARg transcript and proteins expression by phthalates (Figure 2). On Figure 2, the authors demonstrated that it is not the case. To their opinion, it is an important demonstration so they decide to keep this figure in the revised version.
~ Comment 8: In figure 1. B. expression of PPARg protein was checked by western blot. Housekeeping protein used for normalization is not mentioned, and data regarding the same is not presented in this manuscript.
Answer 8: As requested by the editor staff of Life journal, the authors proposed supplemental data during the reviewing process. The reviewer 1 could now read the detailed Western Blot process (supplemental material, Figure 1) where PPARg proteins expression is normalized on total protein expression. This protocol is well established and is published by authors’ team and other; and that is why the authors did not use housekeeping protein.
~ Comment 9: Fig.2 -DMSO 2uM is used as a negative control in a proliferation assay. However, there are no media controls used.
Answer 9: Concerning the Figure 2, the results do not illustrate proliferation assay but cell viability after different treatments. In order to lighten the figure, the authors decided to not indicate the level of cell viability found in the media without DMSO. Indeed, the value is the same for media alone or DMSO 2µM media. The authors had; accordingly, add this result in the paragraph 3.2 (line 17 to 19; page 8) to emphasize this point.
~ Comment 10: The western blot performed (Fig. 3) lower panel PPAR-g protein in AV3 cells. The protein blots are missing.
Answer 10: As mention in the answer of comment 8, supplemental Figure 2 is available to propose proteins blot. This was added after request of Editor Staff during the reviewing process and was not available for reviewer 1.
~ Comment 11: Fetal membrane weakening is induced by the combination of events such as oxidative stress, apoptosis, senescence, extracellular matrix degradation. Data on oxidative stress and ECM degradation in the presence of phthalate could have strengthened the manuscript.
Answer 11: The authors agreed with this comment. Nevertheless, the present paper was the first proposed manuscript to establish (i) phthalates disruption of PPARg pathway in fetal membranes; and (ii) phthalates induction of pro-inflammatory process, an important one in fetal membrane rupture. The authors strongly believed that such results are sufficient to support a publication in Life; with this manuscript as first internationally published concept of proof. Nevertheless, they discuss the other process potential disrupted by phthalates in fetal membranes such as oxidative stress, extra-cellular matrix degradation because PPARg is already published to modulate such molecular events.
~ Comment 12: Effects of MEHP exposure on PPAR-g target genes could be helpful.
Answer 12: The authors already demonstrated the effect of MEHP exposure on IL-6 and IL-8 gene, which are PPARg published target genes. In this way, the authors strongly believe that they had already answer this comment.
~ Comment 13: The discussion requires a revision with a specific explanation of their data. The statement on phthalate and PPARg is largely exaggerated. PPARg expression is an agonist response via several upstream mediators of the external stimuli like this compound (MEHP). Lack of phenotypic data, rationale, data presentation makes the conclusion weak.
Answer 13: As answered in the previous comments, the authors clearly demonstrated that MEHP specifically interferes with PPARg transcriptional activity and not its expression. We agreed that PPARg could answer to other external stimuli but the experimental design of the proposed experiments focus on the interaction of a specific ligand rosiglitazone and MEHP supporting the dysregulation of a specific molecular event: the inflammation. The discussion was developed on such hypothesis, which could be supported by previous publications talking about (i) phthalates present in amniotic fluid, (ii) phthalates docking on PPARg structure, and (iii) PPARg anti-inflammatory properties.

Reviewer 2 Report
The research group of Pr. Vincent Sapin, developed an original article with the objective to evaluate the impact of the most active phthalate (MEHP) on the PPAR gamma pathway in two models of fetal membranes (FM): a) human FM of 1st, 2nd, and 3rd trimester from cesarean and vaginal delivery; b) amniotic epithelial cell line AV3. They evaluated the MEHP effect over PPAR gamma gene and protein expression, FM cytotoxicity, FM viability, the transcriptional activity of PPAR gamma, and IL-6 and IL-8 secretion.
The manuscript is well written, with clarity, fluidity, and is understandable. The methodology section is also well described. The bibliography is adequate.
My main concern is about the emphasis that the researchers put on PPAR gamma dysregulation as a critical factor explaining premature rupture of membranes(pPROM). As the authors indicated, there is no evidence about phthalates' role in this pathology, versus preterm birth in which there is a considerably robust body of evidence. In fact, a past article indicated that PPAR gamma gene SNPs are specific for spontaneous PTB with intact membranes and not for pPROM (doi: 10.1371/journal.pone.0108578). Molecular mechanisms of pPROM are tightly intricated and involve sterile or infectious inflammation response and the extracellular matrix remodeling (which is not evaluated herein). Therefore, conclusions of the manuscript must be directed to the MEHP effect on the suppression of the anti-inflammatory activity of PPAR gamma. I recommend using a tempered language for the pPROM hypothesis.
There are other minor comments:
- The abstract and Figure 1C lack the gamma symbol for PPAR gamma.
- The first paragraph of the introduction lacks a beta symbol for IL-1 beta.
- Introduction. I recommend to detail that PPAR gamma dysregulation by MEHP involves PPAR gamma signaling inhibition, as demonstrated previously in the placenta (reference 8). This will help readers to understand the developed experimental design, combining MEHP (which blocks PPAR gamma signaling) and rosiglitazone (as an agonist inducing PPAR gamma signaling).
- Section 2.5.1 and 2.5.2, the 10 exponent for cells/well must be as superscript.
- Figure 1A. VD-A subheading (vaginal delivery amnion) is shown twice. It is not clear if that is the intention or if it is a mistake and one represents VD-C (vaginal delivery choriodecidua) and the other VD-A.
- Figure 1.A and 1.B. I think Amn I represents isolated amniocytes from FM, but this abbreviature is not explained in the figure legend.
- * p-value should not be present in the legends of Figures 2 and 3 because there are no significant differences.
- Figure 3. The legend should say mRNA instead of RNA. RT-qPCR instead of RT PCR.
- Figures 4 and 5. It is commonly used different letters (i.e. a, b, c) to indicate significant differences among groups. Therefore the use of “a” is somehow confusing. I recommend substituting it for another symbol or a bar.
- Figure 4. It lacks < symbol in the figure legends: *p < 0.05, ** p<0.01, *** p<0.001 compared with the control (DMSO).
- Figure 4. Please indicate the comparing group: a p<0.05 compared with the agonist condition (RGT 1uM).
- Figure 5. Please indicate in the figure that the upper panel is for 24 hours and the lower panel is for 48 hours.
- Figure 5. It lacks < symbol in the figure legends and it is wrong the comparing group: *p < 0.05, ** p<0.01, *** p<0.001 compared with LPS.
- Figure 5. It lacks < symbol in the figure legends and it is wrong the comparing group: a p<0.05 compared with the LPS + RGT 1uM?
- Discussion, 2nd paragraph. As I detailed earlier, I think authors must be more cautious in their sentences, like “... the amnion will be less protected by this nuclear receptor at the end of pregnancy, representing the location of a more intense molecular events occuring for FM rupture”. The authors showed in Figure 1C that there is no difference in PPAR gamma expression between ZIM and ZAM neither in amnion nor in choriodecidua. In addition, the authors exclusively demonstrated that MEHP reverted the antiinflammatory activity of RGT/PPAR gamma by blocking the transcriptional activity of this receptor after the stimulus with LPS. The clinical link between rupture of membranes or pPROM and PPAR gamma dysregulation is not demonstrated herein, therefore the language must be tempered and must clarify this is only a proposed hypothesis.
Author Response
Comments from Reviewer 2
The authors thank the reviewer 2 for the positive comments on the manuscript.
~ Comment 1: My main concern is about the emphasis that the researchers put on PPAR gamma dysregulation as a critical factor explaining premature rupture of membranes (pPROM). As the authors indicated, there is no evidence about phthalates' role in this pathology, versus preterm birth in which there is a considerably robust body of evidence. In fact, a past article indicated that PPAR gamma gene SNPs are specific for spontaneous PTB with intact membranes and not for pPROM (doi: 10.1371/journal.pone.0108578). Molecular mechanisms of pPROM are tightly intricated and involve sterile or infectious inflammation response and the extracellular matrix remodeling (which is not evaluated herein). Therefore, conclusions of the manuscript must be directed to the MEHP effect on the suppression of the anti-inflammatory activity of PPAR gamma. I recommend using a tempered language for the pPROM hypothesis.
Answer 1: The authors agreed with reviewer 2 concerning the fact that phthalates are more described in preterm birth than PPROM. The authors link the PPROM with preterm birth because it is well published that PPROM induce 30-40% of preterm birth. Nevertheless, the authors agreed with the general comment recommending using a more tempered language for PPROM hypothesis. In this way, the authors propose a modified title for the paper: “Dysregulation of the amniotic PPARγ pathway by phthalates: modulation the anti-inflammatory activity of PPARγ in human fetal membranes”. A regulation of related references was realized in the introduction paragraph dealing with the phthalates exposure and obstetrical pathologies i.e. premature birth (13) and PPROM (16) (lines 37 and 39, page 2). In the last sentence of the manuscript, the authors also modified as follow: “In conclusion, by negatively interacting with amniotic PPARγ signaling pathway, the phthalates exacerbate the inflammation, one of the process occurring during physiological rupture of fetal membranes. This exacerbation could lead to pathological weakening of membranes found in 30-40% of preterm birth” (lines 17 to 20, page 15).
~ Comment 2: The abstract and Figure 1C lack the gamma symbol for PPAR gamma.
Answer 2: The authors added the g symbol (lines 21, 22, 24, 26 and 27, page 1) and in figure 1C (page 7).
~ Comment 3: The first paragraph of the introduction lacks a beta symbol for IL-1 beta.
Answer 3: The β symbol was added as requested (line 8, page 2).
~ Comment 4: Introduction. I recommend to detail that PPAR gamma dysregulation by MEHP involves PPAR gamma signaling inhibition, as demonstrated previously in the placenta (reference 8). This will help readers to understand the developed experimental design, combining MEHP (which blocks PPAR gamma signaling) and rosiglitazone (as an agonist inducing PPAR gamma signaling).
Answer 4: The authors agreed with the reviewer and detailed, that PPAR gamma dysregulation by MEHP involves PPAR gamma signaling inhibition in placenta area, a sentence was added in the manuscript (lines 45 to 47, page 2): “The authors demonstrated that MEHP could interfere with the docking of specific ligand (rosiglitazone) to PPARγ leading to a decrease of PPARγ transcriptional activity.”
~ Comment 5: Section 2.5.1 and 2.5.2, the 10 exponent for cells/well must be as superscript.
Answer 5: The 10 exponent was added on lines 18 and 29, page 4.
~ Comment 6: Figure 1A. VD-A subheading (vaginal delivery amnion) is shown twice. It is not clear if that is the intention or if it is a mistake and one represents VD-C (vaginal delivery choriodecidua) and the other VD-A.
Answer 6: The authors have, accordingly, replaced VD-A by VD-C to emphasize this point (page 7).
~ Comment 7: Figure 1.A and 1.B. I think Amn I represents isolated amniocytes from FM, but this abbreviature is not explained in the figure legend.
Answer 7: The abbreviation of primary amniocytes was corrected, adding isolated amniocytes from fetal membranes in the legend of Figure 1 (lines 24 page 7 and line 2 page 8)
~ Comment 8: * p-value should not be present in the legends of Figures 2 and 3 because there are no significant differences.
Answer 8: The p-values were deleted as requested (lines 7 page 9 and 10 page 10)
~ Comment 9: Figure 3. The legend should say mRNA instead of RNA. RT-qPCR instead of RT PCR.
Answer 9: The authors agreed with reviewer 2. RNA and RT-PCR were respectively replaced by mRNA and RT-qPCR in Figure 3 (lines 4 and 5 page 10).
~ Comment 10: Figures 4 and 5. It is commonly used different letters (i.e. a, b, c) to indicate significant differences among groups. Therefore the use of “a” is somehow confusing. I recommend substituting it for another symbol or a bar.
Answer 10: The authors agreed with the reviewer’s comment. Indeed an “a” can be confusing that is why the authors change the “a” to an # (line 16 pages 11 and line 8 page 13).
~ Comment 11: Figure 4. It lacks < symbol in the figure legends: *p < 0.05, ** p<0.01, *** p<0.001 compared with the control (DMSO).
Answer 11: The inferior symbol was added line 15 to 16 page 11.
~ Comment 12: Figure 4. Please indicate the comparing group: a p<0.05 compared with the agonist condition (RGT 1uM).
Answer 12: The RGT 1µM was added line 16, page 11.
~ Comment 13: Figure 5. Please indicate in the figure that the upper panel is for 24 hours and the lower panel is for 48 hours.
Answer 13: The authors agreed with this minor comment and have incorporated the suggestion throughout the manuscript. The upper panel and lower panel were added in Figure 5, page 13.
~ Comment 14: Figure 5. It lacks < symbol in the figure legends and it is wrong the comparing group: *p < 0.05, ** p<0.01, *** p<0.001 compared with LPS.
Answer 14: The symbol was added in line 7 page 13 and the comparing group was changed by LPS.
~ Comment 15: Figure 5. It lacks < symbol in the figure legends and it is wrong the comparing group: a p<0.05 compared with the LPS + RGT 1uM?
Answer 15: The authors have incorporated your suggestion throughout the manuscript. The symbol was added and the legend was modified line 8, page 13.
~ Comment 16: Discussion, 2nd paragraph. As I detailed earlier, I think authors must be more cautious in their sentences, like “... the amnion will be less protected by this nuclear receptor at the end of pregnancy, representing the location of a more intense molecular events occurring for FM rupture”. The authors showed in Figure 1C that there is no difference in PPAR gamma expression between ZIM and ZAM neither in amnion nor in choriodecidua. In addition, the authors exclusively demonstrated that MEHP reverted the anti-inflammatory activity of RGT/PPAR gamma by blocking the transcriptional activity of this receptor after the stimulus with LPS. The clinical link between rupture of membranes or pPROM and PPAR gamma dysregulation is not demonstrated herein, therefore the language must be tempered and must clarify this is only a proposed hypothesis.
Answer 16: As discussed previously, the authors agreed with the reviewer 2 to temper their conclusion. They modified the paragraph 2 of the discussion as following (lines 17 to 27, page 14) : “Our study established that PPARg is expressed during the three trimesters in FMs, leading to early and continuous potential interactions with the phthalates during pregnancy. This is more evident for the amnion in direct contact with the amniotic fluid, where the phthalates are accumulating throughout pregnancy; hence, this layer is more likely to suffer from the phthalates’ toxic properties. In addition, amnion express PPARg at lower levels than choriodecidua and could present decreased related PPARγ molecular actions. With a lower expression of PPARg in the third trimester, the amnion will be less protected against inflammatory attacks by this nuclear receptor at the end of pregnancy. The phthalates could then interfere with this specific process which could be exacerbated and take place in global dysregulation of molecular events (oxidative stress, ECM degradation, senescence…) leading to earlier and more intense weakening of FM. As exposed previously, the authors also temperate their text at the end of the manuscript (lines 17 to 20, page 15).

Reviewer 3 Report
Environmental pollutants represent a health risk that includes the increased incidence of pregnancy complications. The manuscript describes research exploring the effects of a plastic-derived pollutant on the PPAR anti-inflammatory pathway in the human gestational tissues. The focus is the amnion membrane, where pathological inflammation may promote early rupture resulting in premature birth. The Authors address an important issue with a direct link to a clinical problem. Moreover, exposure to pollution is a modifiable risk factor, which makes it urgent and translatable to clarify underlying mechanisms and identify ways of intervention. A further strength of the study is that a variety of experimental approaches and analytical techniques were used to generate outcome measures relevant to PPAR expression, function, and relationship to fetal membrane integrity.
Critical comments:
One major issue is the choice of the AV3 (CCL-21) cell line as the amniocyte model. This cell line is known to be problematic, a HeLa cell derivative (see: https://web.expasy.org/cellosaurus/CVCL_1904 ). The supplier (ATCC) states that “ATCC strongly advises against the use of this cell line as a model for original source material.” Consequently, results generated with these cells can not be considered with confidence to be relevant to amnion physiology and pathophysiology. Primary cells or other amnion cell lines properly validated are informative models of the amnion. The Authors might consider alternatives such replacing the AV3 model to validate key findings, or at least acknowledge this limitation and modify their conclusions accordingly.
A second important issue is the relationship of the MEHP concentration range (1-100 µM) in the cell culture experiments to levels measured in the amniotic fluid. Data available in the literature quite consistently suggest that the amniotic fluid concentrations of this pollutant is historically in the sub-micromolar range. To include this physiologically justified range, at least one order of magnitude lower MEHP concentrations could have been chosen. The pharmacological concentrations are effective as shown, but the interpretation is different.
Minor comments:
The Discussion could be extended to the possibility that MEHP might act by interfering with steroid hormone pathways in the AV3 (endocervical carcinoma-like?) cells.
Page 3, Section 2.1. “Charcoal STRIPPED fetal bovine serum (FBS)” seems correct
Page 4, Section 2.5.1. “3.105 cells/well” – should be 3,105 cells/cell, or similar
Page 4, Section 2.5.2. As described, the antagonist-agonist treatment sequence has been performed twice. Please provide a rationale for this treatment protocol.
Page 5, Section 2.9: “with PPAR-gamma ANTIBODY (1/1000)” seems correct
Page 6, Section 3.1: Please describe how primary amniocytes were prepared/cultured.
Figure 1B: Please indicate trimester
Figure 1C Right Panel: Please indicate trimester and labour status
Figure 2: No need to indicate all significance levels in the legend because no significant difference is reported.
Figure 4 bottom line of legend: Correct to: “a; p<0.05 vs. RGT” (if this is the case).
Figure 5: The no-LPS control is missing and inflammatory activation cannot be assessed (all results are relative to LPS).
Author Response
Comments from Reviewer 3
The authors thank the third reviewer for the positive and constructive comments.
~ Comment 1: One major issue is the choice of the AV3 (CCL-21) cell line as the amniocyte model. This cell line is known to be problematic, a HeLa cell derivative (see: https://web.expasy. org/cellosaurus/CVCL_1904). The supplier (ATCC) states that “ATCC strongly advises against the use of this cell line as a model for original source material.” Consequently, results generated with these cells cannot be considered with confidence to be relevant to amnion physiology and pathophysiology. Primary cells or other amnion cell lines properly validated are informative models of the amnion. The Authors might consider alternatives such replacing the AV3 model to validate key findings, or at least acknowledge this limitation and modify their conclusions accordingly.
Answer 1: This point was already mentioned by the first reviewer and answered by the authors. This cell line has been derived from amniotic cell as stipulated by ATCC catalogue. This cell line is usually used in published papers dealing with amniotic context. For example the authors’ team recently published a paper to study the physiological TLR4 regulation in human FM as an explicative mechanism of pathological preterm case (Belville et. al, e-Life 2022). Another interesting characteristic of this cell line is that it express naturally PPARg, excluding transfection of PPARg plasmid expression. In addition, the use of such model (AV3) could also contribute to the homogeneity of the obtained results compared to the heterogeneity of the results obtained in several primary cells with individual different genetic backgrounds. Nevertheless, as requested by the reviewer, the authors acknowledge this limitation and add the sentence line 15 to 16, page 15.
~ Comment 2: A second important issue is the relationship of the MEHP concentration range (1-100 µM) in the cell culture experiments to levels measured in the amniotic fluid. Data available in the literature quite consistently suggest that the amniotic fluid concentrations of this pollutant is historically in the sub-micromolar range. To include this physiologically justified range, at least one order of magnitude lower MEHP concentrations could have been chosen. The pharmacological concentrations are effective as shown, but the interpretation is different.
Answer 2: The authors agreed with the general comment of the reviewer. Neverethless, the authors propose an experimental plan with MEHP concentration ranges from 1 to 100 µM (i) described as such levels in amniotic fluid and (ii) used in previous papers related to obstetrical environment. In the most recent paper, Shoaïto et al. tested the sub-micromolar concentration without observed effects. It’s why the authors choose to not use this concentration. Others oldest papers focused on placental environment propose 1 to 500 µM concentration ranges. As the reviewer, the authors considered that the concentrations higher than 100 µM are pharmacological ones. By using 1 to 100 µM the authors proposed physiological and active doses in fetal membranes.
~ Comment 3: The Discussion could be extended to the possibility that MEHP might act by interfering with steroid hormone pathways in the AV3 (endocervical carcinoma-like?) cells.
Answer 3: The authors agreed with the third reviewer that MEHP could interfered with steroid hormones receptors as previously published. Nevertheless, the authors have demonstrated that rosiglitazone; a well-established agonist of PPARg, induced an increase in PPARγ transcriptional activity (Figure 4) which is inhibited by MEHP. If this effect was mediated by steroid hormone pathway, the authors would have observed a modification of PPARy expression by MEHP alone, which was not the case. Indeed MEHP alone did not modify the expression of PPARg (Figure 3). So these results show a direct dysregulation of PPARg by MEHP, in terms of transcriptional activity.It is why the authors do not agreed to include this point in the revised version.
~Comment 4: Page 3, Section 2.1. “Charcoal STRIPPED fetal bovine serum (FBS)” seems correct
Answer 4: Stripped (line 15, page 3) was added because, as well noted by the reviewers, the authors always used such type of fetal bovine serum when they designed experimental assay on nuclear receptor with lipophilic ligands.
~ Comment 5: Page 4, Section 2.5.1. “3.105 cells/well” – should be 3,105 cells/cell, or similar
Answer 5: The authors agreed with this and incorporated the suggestion throughout the manuscript. The 10 exponent was corrected (modification asked by the second reviewer) on lines 18 and 29, page 4.
~ Comment 6: Page 4, Section 2.5.2. As described, the antagonist-agonist treatment sequence has been performed twice. Please provide a rationale for this treatment protocol.
Answer 6: In a LPS-induced inflammatory environment, different previously published studies indicated that the cells need to be pretreated by the compound of interest. The incubation time is described from 6 hours (doi: 10.18632/oncotarget.16719) to one day (https://doi.org/10.1155/2015/854359) before the addition of LPS. So the authors decide to pretreat the cells with the agonist 12 hours before the addition of LPS. Moreover, the cells are pretreated 3 hours with MEHP and antagonist reagent before agonist in order to block the PPARg receptor with these compounds. So after the agonist treatment, the cells were incubated with MEHP and GW9662 for 15 hours which allows time for the cells to degrade these compounds (day 1). In order to see the effect of MEHP, the molecule of interest, the authors decided to retreat the cells with these compounds before the induction of inflammatory environment (day 2).
~ Comment 7: Page 5, Section 2.9: “with PPAR-gamma ANTIBODY (1/1000)” seems correct
Answer 7: Antibody was added as requested line 34, page 5.
~ Comment 8: Page 6, Section 3.1: Please describe how primary amniocytes were prepared/cultured.
Answer 8: The protocol of preparation and culture of primary amniocytes was added has requested in paragraph 2.3 (lines 42 to 49, page 3)
~ Comment 9: Figure 1B: Please indicate trimester
Answer 9: The term (3rd trimester) was added in the legend of the figure 1B to emphasize this point line 1 to 2, page 8.
~Comment 10: Figure 1C Right Panel: Please indicate trimester and labor status
Answer 10: The trimester were already indicated in the legend of the figure 1C. The authors added the labor status: caesarian delivery in the legend of the new revised manuscript (line 5, page 8.).
~ Comment 11: Figure 2: No need to indicate all significance levels in the legend because no significant difference is reported.
Answer 11: As also requested by second reviewer, the statistical items were deleted due to the absence of statistical differences.
~ Comment 12: Figure 4 bottom line of legend: Correct to: “a; p<0.05 vs. RGT” (if this is the case).
Answer 12: The legend was corrected (also asked by reviewer number 2) to emphasize this point.
~ Comment 13: Figure 5: The no-LPS control is missing and inflammatory activation cannot be assessed (all results are relative to LPS).
Answer 13: The authors wanted to focus the message of the Figure 5 on the impact of agonist/antagonist and MEHP concerning the reduction of LPS action. It is why they decided to not propose the no-LPS control. In order to take in account the comment of the reviewer, they added in the revised version the follow cytokine expression observed after LPS treatment compared to no-LPS control (line 22 page 11).

Round 2
Reviewer 1 Report
The revised version of the article "Dysregulation of the amniotic PPARγ pathway by phthalates: 2 modulation of the anti-inflammatory activity of PPARγ in human fetal membranes" did not improve much in data visibility, quality presentation and explanation.
The author argues that MEHP does not modulate PPARg transcripts and proteins expression; therefore, determining MEHP levels in these membranes is considered unnecessary. However, the revised article title suggests "Dysregulation of the amniotic PPARγ pathway by phthalates....."
It is not clear how these two arguments co-exist.
Moreover, the author must disclose limitations in the discussion.
Authors remained selective in showing their data, which raised concern about such a selective data exhibition.
At least, on three counts, authors described that data is not shown are much more critical than no-change cell viability data—first, GC analysis of DEHP and Second, comparative DMSO and media control data. Fig. 3 protein blots must be presented.
The author should present or cite a reference that supports that cell lines express PPARg naturally, excluding transfection of PPARg plasmid expression.
There is no data available on phthalates docking on the PPARg structure. Modelling and docking is a different approach involving experimentation and in silco analysis. In the absence of data, it could be misleading to include in the discussion.
The author demonstrated in Figure 2 that there are no changes in cell viability or cytotoxicity irrespective of concentration and duration compared to DMSO.
In that case, subsequent data can be presented with a single concentration (let's assume with 100uM). Data presentation would be better to visualize and understand for the readers.
Check the usage of the word "like against inflammatory attacks."
Authors must work on data presentation and develop a discrete way. Overall, a serious revision can rescue the work.
Author Response
Comments from Reviewer 1
~ Comment 1: The revised version of the article "Dysregulation of the amniotic PPARγ pathway by phthalates: 2 modulation of the anti-inflammatory activity of PPARγ in human fetal membranes" did not improve much in data visibility, quality presentation and explanation.
Answer 1: The authors noted that the first revisions included in the R1 version are not enough in terms of improvements for the reviewer. They answered point by point at all his comments in order to improve the quality and the data visibility.
~ Comment 2: The author argues that MEHP does not modulate PPARγ transcripts and proteins expression; therefore, determining MEHP levels in these membranes is considered unnecessary. However, the revised article title suggests "Dysregulation of the amniotic PPARγ pathway by phthalates...». It is not clear how these two arguments co-exist.
Answer 2: The authors want to thank reviewer 1 to give the opportunity to explain again this point. In this paper, the authors’ aim is to demonstrate the dysregulation of the PPARγ pathway by the phthalates The alteration of signaling supported by nuclear receptors (as usually accepted and published) could occur both at the quantitative level by dysregulating the actor’s concentration (transcript and proteins) and at the qualitative level by disrupting the actor’s activity (transcriptional activity). In this paper, the authors demonstrated that MEHP is able to modulate PPARy activity demonstrating that phthalate can dysregulate this receptor pathway, without modifying the concentration of PPARy transcripts and proteins. Therefore, the obtained results are classical and are not contradictory. As they demonstrated that phthalates are not able to disrupt directly the quantity of PPARy, but more the ability to transduce the signal (as already demonstrated by previous published papers), the determination of phthalates levels are not necessary in the fetal membranes used to determine the quantity of PPARy transcripts and proteins.
~ Comment 3: Moreover, the author must disclose limitations in the discussion.
Answer 3: The authors developed the limitations of the model used for the experiment, which is an immortalized cell line and not primary amniocytes. The authors added this limitation in the conclusion page 15 line 16 to 27. The reviewer 3 also requested this point.
~ Comment 4: Authors remained selective in showing their data, which raised concern about such a selective data exhibition. At least, on three counts, authors described that data is not shown are much more critical than no-change cell viability data—first, GC analysis of DEHP and Second, comparative DMSO and media control data. Fig. 3 protein blots must be presented.
Answer 4: As requested by the reviewer 1 concerning their selectivity in showing their data, the authors decided to add in the Figure 2, 3 and 4 the non-treated cells condition (empty media) to shown that there is no difference with the control condition represent by DMSO. The reviewer 1 could understand why the authors decided in the previous versions to not show this similar result. Moreover, the reviewer 1 can find the protein blots on supplementary material, as requested by the Editorial staff after the submission of the initial version. The blots show the expression of PPARy with no treated cell, DMSO control and all the different treatments illustrated in the related figures.
~ Comment 5: The author should present or cite a reference that supports that cell lines express PPARg naturally, excluding transfection of PPARg plasmid expression.
Answer 5: The AV3 cell line natural expression of PPARy is an information, which cannot really be provided by publication. The cell line was obtain directly by ATCC, which did not propose a systematic global expression pattern of specific proteins, but only few markers to characterize the cell type. However, the authors analyze the endogenous expression of the receptor by RT-PCR and Western blot. The results demonstrated that AV3 cell line express PPARy transcripts and proteins. It is why the authors’ choose a cell line, which endogenously express which is more interesting and simple than choosing a cell line, which need to be transfected by PPARγ expression plasmid. A similar experimental strategy was recently proposed (Shoaito H et al., 2019) with placental trophoblastic cells expressing naturally PPARy without complementary transfection of PPARy plasmid expression. This reference was already present in initial and first revised version.
~ Comment 6: There is no data available on phthalates docking on the PPARg structure. Modelling and docking is a different approach involving experimentation and in silco analysis. In the absence of data, it could be misleading to include in the discussion.
Answer 6: We agreed with reviewer number 1 that modelling and docking experiment are not the same. Several docking studies show that MEHP is able to be fixe in the ligand pocket of PPARy receptor. Moreover, several aspects of the interaction between phthalates and the three PPAR isotypes (α, γ and δ/β) have been studied, including molecular modelling of phthalate. In fact, the ligand binding domains (LBD) of PPARs are well characterized, thus receptor binding of phthalates can now be studied not only in “classic” competitive receptor binding studies, but also by computer based modelling. These studies are based on calculations of the binding energies between the LBD of the PPAR and knowledge of the molecular structure of the respective phthalates (Nakagawa et al., 2008; Kambia et al., 2008; Kaya et al., 2006; Feige et al., 2007). They revealed that the primary metabolites of phthalates (hydrolysed compounds) are more likely PPAR ligands than the parent compounds and the secondary metabolites (oxidised phthalates). For instance, computational studies suggest that MEHP can bind to PPARα and γ, whereas the parent compound DEHP is unable to bind or exhibits weak binding (Feige et al., 2007; Kambia et al., 2008). In accordance with the molecular modelling studies, cell transfection studies report that both mouse and human PPARα and PPARγ were activated by MEHP, but not by the parent compound DEHP (Maloney and Waxman, 1999; Lapinskas et al., 2005). Both type of study (in silico and experimental) have been published about the interaction and impact of phthalate, and MEHP, on the receptor PPARy. Due to this extensive literature on such scientific question (independent of cellular environment), it was not necessary to demonstrate again this point, explaining the absence of such experiments in the proposed manauscript.
~ Comment 7: The author demonstrated in Figure 2 that there are no changes in cell viability or cytotoxicity irrespective of concentration and duration compared to DMSO. In that case, subsequent data can be presented with a single concentration (let's assume with 100uM). Data presentation would be better to visualize and understand for the readers.
Answer 7: The authors really want to show that different range of MEHP, which could interfere, or not with PPARγ pathway. This work allows to see the ‘’global” idea of MEHP impact with low and high doses. It’s why the authors tested (as classically proposed for such of compounds) different concentrations. As the results are different for the dysregulation of PPARy, it was scientifically logical to also test these different concentrations in the other experiences. It’s why we want to maintain a global and homogeneous presentation of the results including the 3 concentrations.
~ Comment 8: Check the usage of the word "like against inflammatory attacks."
Answer 8: The authors thank the reviewer 1 for his comment. The authors agreed with the reviewer to change the expression inflammatory attacks by inflammation page 14 line 24.
~ Comment 9: Authors must work on data presentation and develop a discrete way. Overall, a serious revision can rescue the work.
Answer 9: The authors changed the data presentation by adding the non-treated cells in the Figure 2, 3 and 4. Moreover, the conclusion was tempered, so the authors think that reviewer 1 will be pleased by this new revised version.

Reviewer 3 Report
Regarding Comment 1: The added sentence (page 15, lines 15-16) does not acknowledge the fact that the AV3 cell line is not a suitable model for amnion cells according to evidence and advice from the supplier. The conclusions of the study have not been revised accordingly and/or validation data have not been added.
Regarding Comment 8: The added paragraph (page 3, lines 42-49) describes the procedure of fetal membrane explant cultures, not the procedure to obtain primary amniocytes from the fetal membranes.
Author Response
Comments from Reviewer 3
The authors thank the third reviewer for the constructive comments.
~ Comment 1: The added sentence (page 15, lines 15-16) does not acknowledge the fact that the AV3 cell line is not a suitable model for amnion cells according to evidence and advice from the supplier. The conclusions of the study have not been revised accordingly and/or validation data have not been added.
Answer 1: The authors want to thank reviewer 3 for his comment. The authors decided, according the comment, to change the end of the conclusion. They discussed about the AV3 cell line as a model for fetal membranes, explaining (i) that the cells were transformed with HELA cells, but (ii) that they expressed endogenously PPARγ excluding PPARγ plasmid expression; and (iii) that results need to be confirmed by studying inflammation on primary cells (page 15 line 14 to 23). The following paragraph was inserted “In this study, a cell line model (AV3) was used to assess the impact of MEHP on PPARγ amniotic pathway. Even if it is originally obtained using epithelial amniotic cells, ATCC supplier, indicated that this cell line is contaminated by HELA cells. Nevertheless, this cell line has been characterized for use as a model for investigation of prostaglandin–cytokine interactions in human amnion. AV3 cell line present the same basal levels of IL-6 and IL-8 than those observed in primary amnion cells which is why this cell line was used in this study [29].In addition, we demonstrated that AV3 cell line expressed naturally PPARγ, excluding the transfection of PPARγ plasmid expression leading to a more physiological model of amniotic epithelial cells. The first results need to be comforted by the used of primary amniotic cells » Moreover, the authors tempered they conclusion, indicated that “phthalates could exacerbate the inflammation” in fetal membrane line 22 page 15.
~ Comment 2: The added paragraph (page 3, lines 42-49) describes the procedure of fetal membrane explant cultures, not the procedure to obtain primary amniocytes from the fetal membranes.
Answer 2: The authors want to apologize for the answer proposed previously and they changed the protocol of tissue collection. The authors added the protocol for primary amniocytes obtainment page 3 line 44 to page 4 line 2. The following paragraph was added: “The isolation of human primary amniocytes was conducted in three trypsinization steps (10, 20, and 30 min; trypsin-EDTA 0.25%; 11560626; Gibco) at 37°C, followed by scraping of the amnion. Cells were filtered to remove the collagen, centrifuged for 5 min at 1000 rpm, and grown in T75 flasks coated with collagen type I (04902; Stem Cell Technologies). The amniocytes were cultivated under standard conditions (5% CO2, 95% humidified air, 37°C) in complete Dulbecco modified eagle medium F-12 nutrient mixture supplemented with 10% fetal bovine serum, 100 μg/mL streptomycin, 100 IU/mL penicillin, and 250 μg/mL amphotericin B.”

Round 3
Reviewer 1 Report
The revised version of the manuscript "Dysregulation of the amniotic PPARγ pathway by phthalates: modulation of the anti-inflammatory activity of PPARγ in human fetal membranes" has been improved in rationalizing the author's action. It has added data on primary culture, which has improved the quality of the data output.
The authors need to focus on data presentation, statistical comparison, figure legends, and labelling. Abbreviations used in individual figures should be elaborated within the figure legend.
Fig.1 in the box plot, "+" symbols are visible. It is not clear from the figure legend what it is. The labelling with letters like "A" and "C" conflicts using similar letters in Fig.1 legend.
Fig.1 C unit of mRNA expression? It reads that the authors used multiple housekeeping genes in normalizing the data. It should be mentioned in the figure legend how mRNA expression is normalized with multiple housekeeping genes. Where are the X-axis labels?
Replace “fold change” with “fold expression”.
Statistical analysis suggests multiple comparisons were made among the experimental groups. It is not visible from Fig.1C. The significance (*) is shown between the two groups.
Authors must consult reference papers the way multiple group comparison data are expressed. Like or unlike letters are primarily used to define significance among multiple groups.
Author Response
Comments from Reviewer 1
~ Comment 1: The revised version of the manuscript "Dysregulation of the amniotic PPARγ pathway by phthalates: modulation of the anti-inflammatory activity of PPARγ in human fetal membranes" has been improved in rationalizing the author's action. It has added data on primary culture, which has improved the quality of the data output.
Answer 1: The authors thank the reviewer 1 for the constructive comments and for the help to raise the paper level.
~ Comment 2: The authors need to focus on data presentation, statistical comparison, figure legends, and labelling. Abbreviations used in individual figures should be elaborated within the figure legend.
Answer 2: The authors changed the legend of their data presentation and completed the statistical comparison in Figure 1C (lines 9 to 12; page 8). They also modified the significance representation of multiple comparison of the Figure 4 (line 9; page 11) and 5 (line7; page 13) as requested after.
~ Comment 3: Fig.1 in the box plot, "+" symbols are visible. It is not clear from the figure legend what it is. The labelling with letters like "A" and "C" conflicts using similar letters in Fig.1 legend.
Answer 3: The authors added the meaning of the “+” symbol which represents the mean. The symbol is now explained in the legend of the Figure 1(line 13; page 8). This was similarly done for Figure 2 (lines 8 & 9; page 9), 3 (line 3; page 11), 4 (lines 19& 20; page 11) and 5 (lines 7 and 8; page 11). Concerning the labelling of the figure, the authors already explained in the legend that A is for amnion and C for chorion and do not think that the labelling of the Figure 1A and 1C can be misinterpreted.
~ Comment 4: Fig.1 C unit of mRNA expression? It reads that the authors used multiple housekeeping genes in normalizing the data. It should be mentioned in the figure legend how mRNA expression is normalized with multiple housekeeping genes. Where are the X-axis labels?
Answer 4: The Figure 1 C graphic does not have any unit. In fact, the authors present the results as a fold change between the treatment and the control. The authors added in the legend the method to normalize the data by using housekeeping genes (lines 10 & 11; page 8). The X-axis has been added in the Figure 1 (page 7).
~ Comment 5: Replace “fold change” with “fold expression”.
Answer 5: The “fold change” has been replaced by “fold expression” as request by reviewer 1, in all the concerned figures (see page 7, 9, 10, 11 and 13).
~ Comment 6: Statistical analysis suggests multiple comparisons were made among the experimental groups. It is not visible from Fig.1C. The significance (*) is shown between the two groups.
Answer 6: The authors want to thank the reviewer 1 to pointing this out. The statistic comparison has been added to the legend of the Figure 1C (line 13; page 8). Moreover, the representation of the statistical significance has been modified.
~ Comment 7: Authors must consult reference papers the way multiple group comparison data are expressed. Like or unlike letters are primarily used to define significance among multiple groups.
Answer 7: The authors thank the reviewer 1 for pinpoint this out. Indeed most of the reference papers used letter for multiple group comparison so the authors changed the Figure 1C, 4 and 5 as requested (see page 7, 11 and 13).
Reviewer 3 Report
The added paragraph acknowledges the limitation of the study using the AV3 cell line. The prospective readers have now been provided with the information to interpret the findings in the appropriate perspective.
The inclusion of the cell isolation protocol ensures that the experiment can be reproduced independently by those interested.
Author Response
Comments from Reviewer 3
~ Comment 1: The added paragraph acknowledges the limitation of the study using the AV3 cell line. The prospective readers have now been provided with the information to interpret the findings in the appropriate perspective. The inclusion of the cell isolation protocol ensures that the experiment can be reproduced independently by those interested.
Answer 1: The authors thank the reviewer 3 for the constructive comments and for the help to raise the paper level. Indeed, as indicate by the reviewer, the addition of the cell model’s limitation and the cell isolation protocol improve the paper.